# LSSmScarlet, dCyRFP2s, dCyOFP2s and CRISPRed2s, Genetically Encoded Red Fluorescent Proteins with a Large Stokes Shift

**DOI:** 10.3390/ijms222312887

**Published:** 2021-11-28

**Authors:** Oksana M. Subach, Anna V. Vlaskina, Yuliya K. Agapova, Pavel V. Dorovatovskii, Alena Y. Nikolaeva, Olga I. Ivashkina, Vladimir O. Popov, Kiryl D. Piatkevich, Maria G. Khrenova, Tatiana A. Smirnova, Konstantin M. Boyko, Fedor V. Subach

**Affiliations:** 1Complex of NBICS Technologies, National Research Center “Kurchatov Institute”, 123182 Moscow, Russia; Subach_OM@nrcki.ru (O.M.S.); vlaskina_av@nrcki.ru (A.V.V.); agapova.jk@gmail.com (Y.K.A.); Dorovatovskiy_PV@nrcki.ru (P.V.D.); Nikolaeva_AY@nrcki.ru (A.Y.N.); ivashkina_oi@nrcki.ru (O.I.I.); vpopov@fbras.ru (V.O.P.); 2Bach Institute of Biochemistry, Research Centre of Biotechnology of the Russian Academy of Sciences, 119071 Moscow, Russia; mkhrenova@lcc.chem.msu.ru (M.G.K.); kmb@inbi.ras.ru (K.M.B.); 3Laboratory for Neurobiology of Memory, P.K. Anokhin Research Institute of Normal Physiology, 125315 Moscow, Russia; 4Institute for Advanced Brain Studies, M.V. Lomonosov Moscow State University, 119991 Moscow, Russia; 5Faculty of Biology, M.V. Lomonosov Moscow State University, 119991 Moscow, Russia; 6School of Life Sciences, Westlake University, Hangzhou 310024, China; kiryl.piatkevich@westlake.edu.cn; 7Westlake Laboratory of Life Sciences and Biomedicine, Hangzhou 310024, China; 8Institute of Basic Medical Sciences, Westlake Institute for Advanced Study, Hangzhou 310024, China; 9Department of Chemistry, Lomonosov Moscow State University, 119991 Moscow, Russia; 10Center for Precision Genome Editing and Genetic Technologies for Biomedicine, Institute of Gene Biology, Russian Academy of Sciences, 119334 Moscow, Russia; smirnovatatiana@mail.ru

**Keywords:** genetically encoded red fluorescent proteins, protein engineering, fluorescence imaging, large Stokes shift, LSSmScarlet, dCyRFP2s, dCyOFP2s, CRISPRed2s, red fluorescent, fluorescent protein

## Abstract

Genetically encoded red fluorescent proteins with a large Stokes shift (LSSRFPs) can be efficiently co-excited with common green FPs both under single- and two-photon microscopy, thus enabling dual-color imaging using a single laser. Recent progress in protein development resulted in a great variety of novel LSSRFPs; however, the selection of the right LSSRFP for a given application is hampered by the lack of a side-by-side comparison of the LSSRFPs’ performance. In this study, we employed rational design and random mutagenesis to convert conventional bright RFP mScarlet into LSSRFP, called LSSmScarlet, characterized by excitation/emission maxima at 470/598 nm. In addition, we utilized the previously reported LSSRFPs mCyRFP1, CyOFP1, and mCRISPRed as templates for directed molecular evolution to develop their optimized versions, called dCyRFP2s, dCyOFP2s and CRISPRed2s. We performed a quantitative assessment of the developed LSSRFPs and their precursors in vitro on purified proteins and compared their brightness at 488 nm excitation in the mammalian cells. The monomeric LSSmScarlet protein was successfully utilized for the confocal imaging of the structural proteins in live mammalian cells and multicolor confocal imaging in conjugation with other FPs. LSSmScarlet was successfully applied for dual-color two-photon imaging in live mammalian cells. We also solved the X-ray structure of the LSSmScarlet protein at the resolution of 1.4 Å that revealed a hydrogen bond network supporting excited-state proton transfer (ESPT). Quantum mechanics/molecular mechanics molecular dynamic simulations confirmed the ESPT mechanism of a large Stokes shift. Structure-guided mutagenesis revealed the role of R198 residue in ESPT that allowed us to generate a variant with improved pH stability. Finally, we showed that LSSmScarlet protein is not appropriate for STED microscopy as a consequence of LSSRed-to-Red photoconversion with high-power 775 nm depletion light.

## 1. Introduction

Genetically encoded red fluorescent proteins with a large Stokes shift (LSSRFPs) can be efficiently excited by blue or cyan light while emitting red fluorescence, thus characterized by more than a 100 nm difference between excitation and emission maxima. The large Stokes shift in RFPs is induced by deprotonation of the chromophore in the excited state supported by a hydrogen bond network formed between tyrosine hydroxyl of the chromophore and side chains of the Glu or Asp amino acids located in the close vicinity. The chromophore deprotonation during excitation, also known as excited-state proton transfer (ESPT), has been studied for several LSSRFPs using structural, spectroscopic and mutagenic approaches [1]. The LSSRFPs provide an additional fluorescence color for spectrally multiplexed fluorescence microscopy and are applied for multicolor visualization of the processes in living cells [2]. Due to efficient excitation at 488 nm and 960 nm under one-photon and two-photon microscopy, respectively, LSSRFPs can be simultaneously imaged with green FPs and indicators using dual-emission under common one- and two-photon lasers [3]. Multiple LSSRFPs were reported to date and among them, CyOFP1, CyRFP1, mCyRFP1 and mCRISPRed LSSRFPs stand out in terms of molecular brightness at a 488 nm excitation (Appendix A). However, a direct side-by-side comparison for these proteins is not available, thus creating a need for end-users to test several LSSRFPs to select an optimal variant for a given application. Moreover, it would be desirable to make brighter LSSRFPs using both the brightest RFP as a template, such as mScarlet, and the brightest LSSRFPs as templates, such as CyOFP1, mCyRFP1 and mCRISPRed.

In this manuscript, we report the development of a set of novel LSSRFPs characterized by efficient excitation at a 488 nm laser line. We compared the brightness of the developed LSSRFPs to their precursors in a solution on purified proteins and in the cytosol of mammalian cells. We engineered a version of the brightest RFP mScarlet, with a large Stokes shift, named LSSmScarlet. Monomeric properties of LSSmScarlet correlated with its good localization in the fusions with the most popular structural proteins in mammalian cells. LSSmScarlet was successfully applied for three-color confocal imaging and single excitation dual-emission two-photon imaging of live mammalian cells. We solved the crystal structure of LSSmScarlet with 1.4 Å resolution and performed quantum mechanics/molecular mechanics molecular dynamic (QM/MM MD) simulations and mutagenesis analyses to elucidate the role of amino acids involved in ESPT. Finally, we applied LSSmScarlet protein for STED microscopy.

## 2. Results and Discussion

### 2.1. Developing Red Fluorescent Proteins with Large Stokes Shift Based on mScarlet, mCyRFP1, CyOFP1 and mCRISPRed Proteins in E. coli

For the development of LSSmScarlet, the mScarlet protein was subjected to one round of rational mutagenesis at amino acid positions 147, 162 and 164 followed by eight rounds of random mutagenesis on the mutants having the LSSRed-like phenotype and screening for the highest brightness achieved upon expression in *E. coli* cells. The positions 147, 162 and 164 (corresponding to the positions 148, 165 and 167 according to EGFP enumeration) for the mScarlet directed mutagenesis were suggested according to the residues found in the key positions responsible for the LSSRed-like phenotype for LSSmKates, LSSmOrange, mKeima and other fluorescent proteins [4,5]. Three overlap libraries with mScarlet/S147D,E/I162X/M164X, mScarlet/S147X/I162D,E/M164X and mScarlet/S147X/I162X/M164D,E mutants (Figure 1) were cloned in fusion with sfGFP protein to assist the folding of mutants [6,7,8]. However, sfGFP was so bright that it leaked into the LSSRed channel (excitation at 480/40 nm and emission at 620/60 nm) and actually hampered the screening procedure. Despite this obstacle, we found that mScarlet/S147D/M164C, mScarlet/S147G/I164E and mScarlet/S147D mutants exhibited the LSSRed-like phenotype, i.e., had red fluorescence under blue excitation. We used a mixture of mutants as a template for the first round of random mutagenesis followed by screening for the highest brightness in *E. coli* (or purified protein). After eight rounds of directed molecular evolution, we selected a final variant mScarlet/T74I/Y84H/K122E/W144L/S147D/F178V/K179R mutant, named LSSmScarlet, characterized by the highest molecular brightness at a 488 nm excitation and the absence of red fluorescence at a 561 nm excitation. LSSmScarlet had seven mutations relative to the mScarlet template; five and two were inside and outside of the β-barrel (Figure 1).

To enhance the brightness of CyOFP1, mCyRFP1 and mCRISPRed LSSRFPs in bacterial cells, we subjected these proteins to one, four and four rounds of random mutagenesis followed by screening for the largest brightness under a fluorescence microscope, respectively. As a result of directed molecular evolution, we selected the CyOFP1/N177S, mCyRFP1/N12S/T31I/E118G/N133Y/K165L/K183R and mCRISPRed/N12Y/R126S/N227D mutants, named dCyOFP2s, dCyRFP2s and CRISPRed2s, respectively (Figure 1 and Appendix A). The dCyOFP2s, dCyRFP2s and CRISPRed2s proteins had one, six and three mutations vs. respective parental proteins, and all mutations found in these proteins were outside of the β-barrel (Figure 1).

### 2.2. In Vitro Characterization of the Purified LSSmScarlet, dCyRFP2s, dCyOFP2s and CRISPRed2s LSSRFPs

First, we characterized the spectral properties and molecular brightness of the developed and progenitor LSSRFPs (Figure 2a,b, and Table 1). LSSmScarlet, dCyRFP2s, dCyOFP2s and CRISPRed2s LSSRFPs had maxima of absorption/excitation/emission in the range of 452–508/464–516/590–598 nm, respectively (Figure 2a,b and Table 1). These maxima were close to the progenitor LSSRFPs (Figure 2a,b, and Table 1). Among developed LSSRFPs, LSSmScarlet and CRISPRed2s had the largest Stokes shifts of 128 and 126 nm, respectively (Table 1). According to the molecular brightness at the maximum of excitation, the developed LSSRFPs were ranked as following dCyOFP2s = dCyRFP2s > LSSmScarlet > CRISPRed2s (Table 1). At a 488 nm excitation, the order of the molecular brightness was similar: dCyOFP2s > dCyRFP2s > LSSmScarlet > CRISPRed2s (Table 1). The difference in brightness at a 488 nm excitation between the brightest dCyOFP2s and dimmest CRISPRed2s was 2.3-fold. Parental mCyRFP1 was 1.33-fold less bright than dCyRFP2s; however, original CyOFP1 and mCRISPRed LSSRFPs were 1.33- and 1.13–fold brighter than dCyOFP2s and CRISPRed2s, respectively. Hence, LSSmScarlet and CRISPRed2s proteins were outstanding in terms of the largest Stokes shift, but dCyOFP2s demonstrated the largest molecular brightness at a 488 nm excitation.

The monomeric state of FP is important for protein labeling [9] and low cytotoxicity [10], so we next characterized the oligomeric state of the developed and parental LSSRFPs using fast protein liquid chromatography (FPLC) (Figure 2c and Appendix A and Table 1). On FPLC, LSSmScarlet, dCyRFP2s, dCyOFP2s and CRISPRed2s eluted as a monomer, dimer, dimer and a mixture of the monomer and tetramer, respectively (Figure 2c and Appendix A). Parental CyOFP1 was also dimer as dCyOFP2s; however, mCyRFP1 and mCRISPRed were monomers in contrast to dimeric dCyRFP2s and monomeric-tetrameric CRISPRed2s (Figure 2c and Table 1). Hence, among developed LSSRFPs, only LSSmScarlet is appropriate for the individual protein labeling.

Since the pH-stability of FP is important for imaging in different cellular organelles with varied pH values (5.0 in lysosomes to 7.5 in the cytosol [11]), we further characterized the dependence of the fluorescence of the developed and parental LSSRFPs on the variation of pH (Figure 2d and Table 1). Except for LSSmScarlet, all other developed LSSRFPs dCyRFP2s, dCyOFP2s and CRISPRed2s demonstrated around a 2-fold drop of their fluorescence at the increasing of the pH in the range of a pH of 7–10 with pK_a_ values of around 10; in turn, the fluorescence of LSSmScarlet was at maximum and did not change within this pH range (Figure 2d). Acidification resulted in the loss of the dCyRFP2s’ and dCyOFP2s’ fluorescence with pKa values of 5.44 and 5.29, respectively (Figure 2d and Table 1). At acidification, the fluorescence of LSSmScarlet dropped by around 2-fold with a pK_a_ value of 5.78 and further diminished completely with a pK_a_ value of 1.91 (Figure 2d and Table 1). At acidic pH values, the fluorescence of CRISPRed2s was characterized by a pK_a_ value of 2.23 (Figure 2d and Table 1). The pH stabilities for the parental LSSRFPs were similar to their derivatives (Figure 2d and Table 1). The shapes of the excitation, emission and absorption spectra for LSSmScarlet protein were similar in the pH range 2.0–11.5 (Appendix A). However, at pH values above 11.5, the LSSmScarlet showed an additional RFP-like anionic form of the chromophore with absorption/excitation/emission peak maxima at 570/570/607 nm, respectively (Appendix A). The spectral properties of this RFP-like anionic form of the LSSmScarlet were similar to absorption/emission maxima for the mScarlet RFP observed at 569/598 nm, respectively [12]. Hence, among all LSSRFPs tested, LSSmScarlet demonstrated the highest pH-stability at alkaline pH values and, together with CRISPRed2s, better preserved its fluorescence at acidic pH values.

We next compared the one-photon photostability of the developed and parental LSSRFPs on purified proteins using microdroplets’ suspension in oil under continuous illumination with a metal halide lamp at a 470/40 nm excitation and 63x oil objective lens (Figure 2e and Table 1). According to the measured photobleaching half-times, the developed LSSRFPs were ranked as dCyOFP2s > dCyRFP2s > LSSmScarlet > CRISPRed2s (Table 1). The difference in photostability between the most photostable dCyOFP2s and the least photostable CRISPRed2s was 2.2-fold. In turn, the photostabilities of the parental LSSRFPs were similar to photostabilities of the developed LSSRFPs (Figure 2e and Table 1). Under identical conditions, the commonly used mEGFP protein had a photostability 1.7–3.9-fold higher as compared to the developed LSSRFPs (Figure 2e and Table 1). Hence, the developed LSSRFPs demonstrated a one-photon photostability 1.7–3.9-fold lower than mEGFP, and the dCyOFP2s protein was the most photostable among them.

We further measured the maturation rate and efficiency for the engineered and original LSSRFPs at 37 °C using purified proteins expressed in bacterial cells (Figure 2f and Table 1). According to the maturation half-times, the developed LSSRFPs were ranked as following dCyOFP2s < dCyRFP2s < LSSmScarlet < CRISPRed2s (Figure 2f and Table 1). The difference in maturation half-times for dCyOFP2s with the fastest maturation and CRISPRed2s with the slowest maturation was 3.8-fold. Parental CyOFP1 showed the same maturation half-time as dCyOFP2s; the mCyRFP1 progenitor matured 1.5-fold faster than dCyRFP2s, and mCRISPRed had a 3.9-fold slower maturation than CRISPRed2s (Figure 2f and Table 1). We also estimated the maturation efficiency of the LSSRFPs according to normalization of the extinction coefficient calculated at 280 nm to the extinction coefficient determined by alkaline denaturation (Appendix A). All LSSRFPs tested revealed a maturation efficiency of 68% and higher. Hence, all developed LSSRFPs matured efficiently, and among all of them, dCyOFP2s exhibited the fastest maturation rate.

### 2.3. Brightness of the LSSmScarlet, dCyRFP2s, dCyOFP2s and CRISPRed2s LSSRFPs in Cultured Mammalian Cells

We next compared the intracellular brightness of the developed and progenitor LSSRFPs in the cytosol of live HeLa cells at a 488 nm excitation light under a confocal microscope (Figure 3 and Table 2). To account for differences in the expression levels, LSSRFPs were co-expressed with the EGFP protein via the self-cleavable P2A peptide linker. All LSSRFPs were evenly distributed in the cytosol and nucleus of the HeLa cells similarly to EGFP (Figure 3a and Appendix A). According to the measured intracellular brightness normalized to the green fluorescence intensity of EGFP, the developed LSSRFPs were ranked as following dCyOFP2s = dCyRFP2s > LSSmScarlet > CRISPRed2s (Figure 3 and Table 2). The difference in brightness between the brightest dCyRFP2s and dimmest CRISPRed2s was 3.4-fold (Figure 3 and Table 2). Parental mCyRFP1 and CyOFP1 were 1.29- and 1.15-fold dimmer than the dCyRFP2s and dCyOFP2s proteins, respectively (Figure 3 and Table 2). The mCRISPRed progenitor did not show a statistically significant difference in brightness as compared to CRISPRed2s (Figure 3 and Table 2). We noticed a difference in brightness of LSSRFPs expressed in HeLa cells and purified from bacteria (Table 2). This difference was the largest for mCRISPRed and CyOFP1 proteins; these proteins were 2.08- and 1.93-fold brighter according to the molecular brightness determined on purified proteins as compared to the brightness in HeLa cells. The observable differences in brightness of FPs expressed in mammalian cells over bacterially expressed and purified proteins might be associated with their folding and maturation efficiency, and/or stability inside mammalian and bacterial cells [13,14,15]. Overall, dCyRFP2s and dCyOFP2s demonstrated the highest brightness in the cytosol of mammalian cells and were 1.29- and 1.15-fold brighter than their mCyRFP1 and CyOFP1 progenitors, respectively.

### 2.4. Behavior of LSSmScarlet in Mammalian Cells in Fusions with Structural Proteins

Since among the developed LSSRFPs, LSSmScarlet was the only one with monomeric behavior, we decided to test it further in mammalian cells in different protein fusions (Figure 4). LSSmScarlet demonstrated right microtubules-like localization in the fusion with β-actin, α-tubulin and vimentin cytoskeleton proteins when expressed in HeLa cells (Figure 4a). The dMito signal properly targeted LSSmScarlet into the lumen of mitochondria (Figure 4a). Fusion of LSSmScarlet with H2B histone protein localized in the nuclei of the cells and did not block cells from division (Figure 4b).

We compared the localization of LSSmScarlet with novel emiRFP703 far-red protein proposed for fusion and subcellular structural imaging [16]. In fusion with vimentin, LSSmScarlet demonstrated better localization than emiRFP703 (Appendix A), since in the case of emiRFP703, we found many cells with obvious mislocalization (Appendix A, bottom). Hence, LSSmScarlet is appropriate for the labeling of the individual proteins in mammalian cells.

### 2.5. Structural Characterization of LSSmScarlet

To understand the molecular mechanism of a large Stokes shift in LSSmScarlet and the impact of mutations introduced during directed molecular evolution, we determined its crystal structure using X-ray diffraction at 1.4 Å resolution (Figure 5 and Table 3). There is one protein chain per asymmetric unit, and a contact analysis revealed that the protein is a monomer in the crystal at pH 6.5, i.e., in a highly fluorescent state (Figure 5a). LSSmScarlet has a typical β-barrel fold with the mCherry-like chromophore formed by ^67^MYG^69^ amino acids and positioned on the central helix of the barrel (Figure 5a,b). The chromophore also forms four direct hydrogen bonds (H-bonds) to R71, R96, D147 and E216, and several water-mediated H-bonds to Q65, Q110, T112, E145 and L200 (Figure 5c,d). The phenolic hydroxyl group of the chromophore forms two hydrogen bonds with the carboxyl group of D147 and with the buried water molecule. The negative charge of the pair D147/phenolic hydroxyl group of the chromophore is maintained by R198 that is stacked with the phenolic group of the chromophore as well as hydrogen-bonded to D147 via the solvent molecule.

To elucidate the mechanism of the excited state proton transfer (ESPT) and a large Stokes shift in LSSmScarlet, we analyzed contacts of the phenolic hydroxyl group of the chromophore with its environment. The structure revealed that the phenolic hydroxyl group of the chromophore is engaged in direct contact with the carboxyl group of D147 with the donor-acceptor distance of 2.56 Å. Since the excitation maximum is blue-shifted by 128 nm relative to the emission maximum in LSSmScarlet compared to only a 25 nm difference between excitation and emission maxima in its predecessor mScarlet, the phenolic hydroxyl group is likely to exist in the protonated form in the ground state. Upon excitation of the protonated chromophore in LSSmScarlet, excited-state proton transfer (ESPT) occurs, wherein the phenolic group of the neutral protonated chromophore serves as a donor of a proton to a negatively charged D147 side chain, and the formed anionic chromophore fluoresces (Figure 6a). Thus, we can explain the observed large Stokes shift in LSSmScarlet by ESPT from the chromophore to the carboxylic group of the D147 side chain (as proved below by molecular simulations). Large Stokes shifts in other LSSFPs were also reported to be explained by ESPT. The resolved crystal structures for CyOFP1 (PDB:5BQL) [3] and mCRISPRed (PDB: 6XWY) [17] demonstrated that ESPT from the phenolic hydroxyl group of the chromophore to the K160 amino group in CyOFP1 (corresponding to M164 in LSSmScarlet) and the D142 carboxyl group in mCRISPRed (corresponding to D147 in LSSmScarlet) underlie large Stokes shifts in these FPs.

To understand the impact of mutations introduced during LSSmScarlet development, we compared chromophore planarity and the chromophore’s environment for LSSmScarlet and the parental protein mScarlet (PDB: 5LK4) [12] (Figure 7). The relative positions of imidazolinone and phenolic rings in the LSSmScarlet and mScarlet chromophores are different. Two rings of the chromophore are almost perfectly in the plane in mScarlet, but phenolic ring is out of the plane in LSSmScarlet with corresponding twist and tilt angles of about 5 and 20° (according to Figure 6 in [4]). This difference in the chromophore planarity results in the higher quantum yield exhibited by mScarlet with a more planar conformation of the chromophore (QY for mScarlet is 0.7 versus 0.43 for LSSmScarlet); similarly, a decrease in quantum yield for mFruits proteins was associated with chromophore noncoplanarity [18]. Less planar conformation of the chromophore in LSSmScarlet could be a result of key mutation S147D necessary for the large Stokes shift. This mutation also led to a strong shift of the M164 side chain (RMSD between corresponding S atoms of mScarlet and LSSmScarlet is 3.64 Å) due to steric hindrance.

There are five mutations in LSSmScarlet compared to mScarlet that are inner to the β-barrel: S147D, W144L, F178V, T74I and Y84H. Three mutations, S147D, W144L and F178V were located at distances in the range of 6 Å away from the chromophore. Because of the strong shift of M164 caused by the S147D mutation, there is a steric clash. The new position of Met164 in LSSmScarlet required the replacement of adjacent bulky aromatic amino acid residues W144 and F178 with smaller ones—Leu and Val, respectively. The former mutation also seems to stabilize the L200 side chain (one conformation in LSSmScarlet compared to two in mScarlet), which forms van der Waals contacts with the chromophore. Mutations, T74I and Y84H, are more than 6.0 Å away from the chromophore; however, T74 substitution breaks the hydrogen bond between T74 and S218, which in turn forms a hydrogen bond with E216. This leads to a reorientation of the S218 side chain towards E216, strengthening the hydrogen bond S218–E216 and a shift of the glutamic acid side chain towards L200 (compared to mScarlet).

### 2.6. Molecular Dynamics Simulations of the ESPT Mechanism in LSSmScarlet

We performed QM/MM MD simulations based on the available X-ray structure of the LSSmScarlet protein to elucidate the molecular mechanism of the proton transfer occurring in the excited electronic state, S_1_ (Figure 6b). The 20 ps ground state (S_0_) QM/MM MD simulations demonstrated that the chromophore exists in the neutral state, and the D147 is negatively charged. First, we performed QM/MM MD on the first excited electronic state, S_1_, starting from the representative frame of the trajectory on the ground state, S_0_. After vertical excitation, the initial relaxation of the conjugated π-system of the neutral chromophore occurs (Figure 6b). It resulted in a small decrease (about 0.2 eV) of the energy gap between the ground and excited states. Next, the proton is transferred from the phenyl fragment of the chromophore to the carboxylate of D147, and this proton transfer results in the formation of the fluorescent state and a considerable decrease in the energy gap between the S_0_ and S_1_ surfaces. According to our model, the Stokes shift is ~0.7 eV that agrees with the experimental 0.57 eV and further proves the suggested ESPT mechanism. Hence, QM/MM MD simulations based on X-ray data for the LSSmScarlet protein proved the ESPT that is responsible for the large Stokes shift red fluorescence in this protein.

### 2.7. The Role of R198 Residue in ESPT of the LSSmScarlet Protein

To elucidate the role of R198 residue in ESPT, we generated five mutants of LSSmScarlet containing R198E, R198H, R198I, R198K and R198Y substitutions and characterized their spectral properties and pH stabilities. LSSmScarlet with R198E and R198H substitutions preserved efficient ESPT of the chromophore at pH 7.40 according to the single absorption peaks at 480–487, corresponding to the protonated chromophore (Table 4 and Figure 8). The R198I mutation resulted in the formation of a deprotonated chromophore at pH 7.40 according to the appearance of the absorption peak at 571 nm (Table 4 and Figure 8). R198K and R198Y mutants did not efficiently form the red chromophore according to the major absorption peaks at 405 and 372 nm, respectively (Table 4 and Figure 8), so we did not characterize them further. LSSRed fluorescence of R198H, R198E and R198I LSSmScarlet mutants showed dependence on pH with pKa values of 2.403, 6.181 and 7.76, respectively (Table 4 and Figure 8). We assumed that H198 and E198 residues served as proton donor/acceptors via the W17 molecule and stabilized ESPT from D147 residue to the chromophore (Figure 5, Figure 6 and Figure 7), but I198 did not. Positively charged H198 residue stabilized ESPT to a higher extent as compared to negatively charged E198 probably because of favorable and unfavorable electrostatic interactions with negatively charged D147 residue, respectively. According to X-ray structures’ superposition, mCRISPRed LSSRed protein with high pH stability also has positively charged H196 residue in a position corresponding to position 198 in LSSmScarlet (Appendix A). However, H196 residue of mCRISPRed did not form the H-bond with D142 residue, which makes a direct H-bond to the chromophore of mCRISPRed (analogous to D147 residue in LSSmScarlet). Probably, an electrostatic interaction between positively charged H196 (R198) and negatively charged D142 (D147) is important for the high pH-stability of LSSRed fluorescence for both proteins. Hence, the residue in position 198 of LSSmScarlet may participate in ESPT to the chromophore as a proton donor/acceptor, and a positive charge in position 198 facilitates ESPT at an acidic pH values.

### 2.8. Two- and Three-Color Confocal Imaging with the LSSmScarlet Protein

We next validated the utility of LSSmScarlet for multi-color confocal imaging in combination with conventional GFPs and RFPs at 488 and 561 nm excitations. First, we verified the bleed-through of LSSmScarlet fluorescence into the standard green and red channels under a confocal microscope. We expressed H2B-LSSmScarlet fusion protein in HeLa cells and imaged it in green, LSSRed and red channels (Figure 9a). We did not see a notable bleed-through of the LSSmScarlet fluorescence into both the green and red channels (Figure 9a). No bleed into the green channel was expected according to the good separation of the LSSmScarlet emission spectra and the green emission filter (Figure 2b). We also anticipated that LSSmScarlet will not bleed into the red channel, because according to its excitation spectrum, it is excited with 561 nm light 27-fold less efficiently compared to 488 nm light (Figure 2b). Moreover, the absence of the bleed into the green channel was estimated and confirmed for other LSSRFP CyOFP1 [3]. Hence, during confocal imaging, the LSSmScarlet protein does not bleed-through into the green and red channels when excited at 488 and 561 nm, respectively, and it should be applicable for combined confocal imaging together with GFP- and RFP-like proteins.

To demonstrate single excitation two-color emission confocal imaging with the LSSmScarlet protein and mNeonGreen-derived novel family of fluorescent calcium indicators [19], we imaged mammalian cells co-expressing the H2B-LSSmScarlet fusion and NES (nuclear export signal)-NCaMP4 green-yellow calcium indicator using a single 488 nm excitation light and standard emission filters. We transiently co-expressed the H2B-LSSmScarlet protein and NES-NCaMP4 green-yellow calcium indicator in the nucleus and cytosol of the HeLa cells, respectively. Confocal imaging with a 488 nm excitation demonstrated that the green-yellow NES-NCaMP4 indicator and LSSRed H2B-LSSmScarlet protein had anticipated localization (Figure 9b). The NES-NCaMP4 revealed a slight bleed-through into the LSSRed channel, which was not seen on the overlaid image (Figure 9b). For the green EGFP protein, the bleed-through into the LSSRed channel should be similar to that for green-yellow NCaMP4 since EGFP has a similar emission spectrum integral in the range of the 617/73 nm red emission filter as compared to the integral for the NCaMP4 calcium indicator (Appendix A). Hence, the LSSmScarlet protein is appropriate for single excitation two-color confocal imaging with GFP- and mNeonGreen-derived fluorescent proteins.

Since the LSSmScarlet protein does not bleed through into the red channel (please, see above), we next attempted to demonstrate three-color confocal microscopy with the LSSmScarlet LSSRFP, mCherry RFP and mNeonGreen-derived green-yellow NeonOxIrr indicator for hydrogen peroxide [20] using a 488/561 nm dual excitation and standard emission filters. We transiently co-expressed dMito-LSSmScarlet, H2B-mCherry and Vimentin-NeonOxIrr fusion proteins in the lumen of mitochondria, in the nucleus and in the cytoskeleton of the HeLa cells. Confocal imaging with a 488 and 561 nm excitation and dual-emission revealed an anticipated localization of the dMito-LSSmScarlet, H2B-mCherry and Vimentin-NeonOxIrr fusion proteins in HeLa cells (Figure 9c). We did not notice any bleed-through between all three channels. Hence, LSSmScarlet protein can be used for three-color confocal imaging with GFP-, mNeonGreen- and mCherry-derived fluorescent proteins.

### 2.9. Two-Photon Imaging of the LSSmScarlet Protein

To assess the applicability of the LSSmScarlet protein for two-photon (2P) fluorescence microscopy, we acquired two-photon images of the HeLa cells expressing H2B-LSSmScarlet and studied the dependence of the 2P excitation of the H2B-LSSmScarlet vs. 2P excitation wavelength. Using 11% of the laser power (of the standard Ti-Sapphire laser family) of the 2P microscope at a 940 nm excitation wavelength, we successfully imaged H2B-LSSmScarlet expressed in HeLa cells (Figure 10a). The maximal fluorescence intensity of the H2B-LSSmScarlet was observed at a 900 nm two-photon excitation wavelength (Figure 10b). The maximum of the H2B-LSSmScarlet fluorescence intensity normalized to the laser power was found at 940 nm (Figure 10b). This maximum wavelength coincided with a doubled one-photon excitation maximum wavelength for the purified LSSmScarlet protein observed at 470 nm (Figure 10b and Table 1). Hence, LSSmScarlet is appropriate for two-photon fluorescence microscopy with a standard Ti-Sapphire laser family, and a 940 nm wavelength is optimal for its 2P excitation.

We next performed 2P imaging of the NES-NCaMP4 green-yellow mNeonGreen-based calcium indicator and tested the dependence of its fluorescence on the 2P-excitation wavelength. Using 10% of the laser power at 940 nm, we visualized the NES-NCaMP4 indicator transiently expressed in the cytosol of the HeLa cells (Figure 10c). The maximal fluorescence for the NES-NCaMP4 indicator was observed at 900 nm (Figure 10d). The maximum of the NES-NCaMP4 fluorescence normalized to the laser power was seen at 940 nm (Figure 10d). This maximum was the same for both NCaMP4 and LSSmScarlet proteins. Hence, the NES-NCaMP4 indicator is appropriate for 2P imaging, and 940 nm is the optimal wavelength for its 2P excitation.

To demonstrate single excitation dual-emission 2P microscopy with LSSmScarlet and the mNeonGreen-based reporter, we imaged H2B-LSSmScarlet protein and the NES-NCaMP4 indicator co-expressed in the nucleus and cytosol of the HeLa cells using 2P microscopy with a single excitation at 940 nm and standard emission filters. Using 10% laser power at a 940 nm excitation, we successfully acquired 2P images of the H2B-LSSmScarlet and NES-NCaMP4 localized in the nucleus and cytosol of the HeLa cells, respectively (Figure 10e). We noted bleed-through of the NCaMP4 fluorescence into the LSSRed channel, but this bleed was weak, and these two reporters were clearly distinguished in the superimposed image (Figure 10e). Simultaneous dual-emission 2P imaging was also earlier demonstrated for another LSSRFP CyOFP1 protein together with GFP protein and the G-CaMP6s calcium indicator in live cells in vitro and in vivo [3]. Hence, the LSSmScarlet protein is appropriate for single excitation dual-emission 2P microscopy with GFP- and mNeonGreen-based fluorescent reporters.

### 2.10. LSSRed-to-Red Photoconversion of the LSSmScarlet Protein in the Conditions of the Super-Resolution STED Imaging

Finally, we assessed whether LSSmScarlet protein is suitable for super-resolution stimulated emission depletion (STED) microscopy similar to mKeima LSSRFP [21].

Since STED imaging is often performed on fixed samples, we studied the impact of fixation on the fluorescence of the LSSmScarlet protein expressed in mammalian cells. Using the confocal microscope, we imaged HeLa cells expressing the H2B-LSSmScarlet fusion protein before and after fixation with 4% polyformaldehyde (PFA) for 15 min at room temperature. We observed minor quenching of the fluorescence upon fixation, which was not statistically significant (*p* = 0.5212; Appendix A). Resistance of the LSSmScarlet protein to the fixation correlated with its high pH-stability.

Using fixed HeLa cells containing vimentin-LSSmScarlet protein, we could not obtain notable improvement in the resolution of the vimentin fibers using a 488 excitation light and both 775 nm (Figure 11a) and 595 nm (Figure 11b) depletion lasers. Depletion by the 775 nm laser did not affect image resolution in a wide range of excitation and depletion laser powers, and we did not succeed in resolution improvement. The 595 nm was probably an inappropriate wavelength for the LSSmScarlet depletion according to its emission spectrum (Figure 2b). We found that the 775 nm depletion light bleached the LSSRed fluorescence of the LSSmScarlet protein excited with 488 nm light, and new red fluorescence at 561 nm excitation appeared (Figure 11c,d).

To study photochemical transformations for the LSSmScarlet protein, we registered spectral changes accompanied by irradiation of the LSSmScarlet purified protein with a two-photon 775 nm laser, and one-photon 405 and 468 nm LED arrays. We illuminated purified LSSmScarlet protein with a 775 nm 2P laser (4010 mW power) through a 25× objective lens and found the formation of the traces of the red form with excitation/emission maxima at 570/598 nm, respectively (Appendix A). We suggest that the same red-form is formed in the conditions of the STED imaging (Figure 11c,d). The one-photon illumination of the LSSmScrlet protein with a 405 nm LED array resulted in the formation of the green-form with an absorption peak at 412 nm and excitation/emission maxima at 426/503 nm, respectively (Figure 10b); the traces of the red-form were also registered according to the excitation/emission peaks at 550/592 nm, respectively (Figure 10b). The one-photon irradiation of the LSSmScarlet protein with the 468 nm LED array resulted in the formation of the cyan-form with an absorption peak at 401 nm and excitation/emission maxima at 422/480 nm, respectively (Figure 10c); compared to 405 nm light, we observed more traces of the red-form with an absorption at 570 nm and excitation/emission maxima at 554/598 nm, respectively (Figure 10c). LSSRed-to-Red photoswitching of the PSLSSmKate protein with a one-photon 405 nm light illumination was earlier attributed to decarboxylation of the glutamate 215 residue via the Kolbe-type reaction mechanism [22]. As compared to PSLSSmKate, the irradiation of the LSSmScarlet protein with 405 and 468 nm light resulted in the formation of the different spectral species. Hence, the LSSmScarlet protein undergoes various photochemical transformations with the formation of the green, cyan and red fluorescent products depending on the light power and the wavelength of light.

Finally, using live HeLa cells expressing vimentin-LSSmScalert fusion, but not the fixed ones, we could perform STED imaging of the Red-photoconverted form of the LSSmScarlet protein with a 561 nm excitation and 775 nm depletion (Figure 11e). As compared to confocal imaging, using STED microscopy, we could improve resolution of vimentin fibers by 2.4-fold (Figure 11e). Overall, the LSSmScarlet protein is not a protein of choice for STED microscopy as a consequence of the photoconversion with a high-power 775 nm depletion laser.

## 3. Materials and Methods

### 3.1. Cloning of Bacterial Vectors, Mutagenesis and Library Screening

LSSRFP proteins were cloned into the pBAD/HisB plasmid (Invitrogen) at BglII/EcoRI restriction sites using the mSc-BglII/mCherry-EcoRI-r (LSSmScarlet), Fw-LSSmOrange-BglII/mCherry-EcoRI-r (CyOFP1), Fw-LSSmOrange-BglII/Rv-LSSmOrange-EcoRI (mCyRFP1) and Fw-BglII-(PA)TagRFP/Rv-LSSmOrange-EcoRI (mCRISPRed) primers listed in Appendix A to express LSSRFPs proteins in BW25113 bacterial cells (kindly provided by Verkhusha V.V. from Albert Einstein College of Medicine, New York, NY, USA).

Three site-directed libraries for rational mutagenesis of the parental mScarlet protein at three positions were generated using mSc-148DE, mSc-148DE-r, mSc-165,7-X, mSc-165,7-X-r, mSc-148X, mSc-148X-r, mSc-165,7-DX, mSc-165,7-DX-r, mSc-165,7-XD and mSc-165,7-XD-r primers listed in Appendix A. Assembly of the whole gene was performed using PCR with overlapping fragments [23]. Generated libraries were inserted at BglII/EcoRI restriction sites of the pBAD/HisB-TorA-HyPer-sfGFP plasmid by swapping the HyPer gene.

Random libraries of LSSRFPs were obtained using PCR in the presence of Mn^2+^ ions (according to the Diversify PCR Random Mutagenesis Kit User Manual, Clontech, 2–3 random mutations were introduced per 1000 base pairs) and cloned at BglII/EcoRI restriction sites of the pBAD/HisB plasmid.

Screening of bacterial libraries was performed on Petri dishes under a fluorescent microscope. Briefly, expression of the LSSRFPs on the colonies on Petri dishes was induced with 0.02% arabinose for 24 h at 37 °C. Screening of about 10,000 colonies of the bacterial library expressing LSSRFPs variants was performed on Petri dishes under fluorescent stereomicroscope Leica M205FA (Leica, Germany) equipped with the DFC310FX camera (Leica Microsystems, Germany) and mercury metal halide light source EL6000 (Leica Microsystems, Germany). LSS-red fluorescence was registered by 480/40BP excitation (62 µW/cm^2^ on the sample) and 620/60BP emission filters. Acquired images were analyzed using ImageJ software, and colonies having the highest brightness were picked up for further analysis on bacterial streaks on Petri dishes.

### 3.2. Proteins’ Purification and Characterization

Proteins were expressed and purified using the pBAD/HisB arabinose-inducible system (Invitrogen) as described in reference [7]. Briefly, BW25113 bacterial cells were transformed with pBAD/HisB-LSSRFPs plasmid. For protein expression, the bacterial cultures were grown in LB medium supplemented with 0.004% arabinose and 100 μg/mL ampicillin overnight at 37 °C and 220 rpm. The cultures were centrifuged at 4648× *g* for 10 min. The cell pellets were resuspended in PBS buffer, pH 7.4, supplemented with 300 mM NaCl and 10 mM Imidazole and lysed by sonication on ice. The sonicated solution was centrifuged at 36,670× *g* at 4 °C. The proteins were further purified using Ni-NTA resin (Qiagen) followed by dialysis for 16 h against PBS buffer.

The extinction coefficients’ values for the purified LSSRFPs proteins were calculated in PBS buffer, pH 7.4, using the alkaline denaturation method and assuming that the GFP-like chromophore has the extinction coefficient of 44,000 M^−1^ cm^−1^ at 455 nm in 1 M NaOH [24]. Absorption spectra were recorded using the NanoDrop 2000c Spectrophotometer (Thermo Scientific, DE, USA).

The quantum yields for purified LSSRFPs proteins excited at 460 nm were measured by a comparison of the integrated fluorescence values (in the range of 470–800 nm) in PBS buffer, pH 7.40, with the similarly integrated fluorescence values for the equally absorbing at 460 nm CyOFP1 protein (quantum yield of 0.76 [3]). Fluorescence spectra were acquired using a CM2203 spectrofluorometer (Solar, Minsk, Belarus).

pH titrations for purified LSSRFPs proteins (32–202 nM final concentration) were performed in buffers of 30 mM citric acid, 30 mM borax, 30 mM NaCl with a pH ranging from 1.5 to 10.0, using incubation for 20 min at room temperature. LSS-red fluorescence (Ex490nm/Em580–640nm) was registered using 96 well ModulusTM II Microplate Reader (Turner Biosystems, Sunnyvale, CA, USA).

Photobleaching of LSSRFPs was performed using suspensions of purified proteins in mineral oil. Briefly, the photobleaching was measured using a suspension of 1.5 µL of purified proteins dialyzed in PBS buffer, at a 45 µM concentration, in 10 µL of mineral oil placed on the cover glass. Imaging of the suspension was performed using the Zeiss Axio Imager Z2 microscope (Zeiss, Germany) equipped with a 120 W mercury short-arc lamp (LEJ, Germany), a 63 × 1.4 NA oil immersion objective lens (PlanApo, Zeiss, Germany), a 470/40BP excitation filter, a FT 495 beam splitter and a 605/70BP emission filter. Light power density (3.78 mW/cm^2^) was measured at a rear focal plane of the objective lens using the PM100D power meter (ThorLabs, Germany) equipped with the S120VS sensor (ThorLabs, Germany). No corrections were applied to the experimental data.

Size-exclusion chromatography was performed with a SuperdexTM 75 10/300 GL column using the GE AKTA Explorer (Amersham Pharmacia, UK) FPLC System.

To assess the maturation rate of LSSRFPs, 20 mL of bacterial cultures were grown in 50 mL tubes with LB medium supplemented with 100 µg/mL ampicillin at 37 °C, 190 rpm, overnight. Next, protein expression was induced by the addition of 0.2% arabinose, and the culture was transferred into a 15 mL tube until the brim. Protein expression lasted for 3 h at 37 °C, 190 rpm at anaerobic conditions. The cultures were then centrifuged at 3500× *g* for 10 min at room temperature. The pellet was resuspended in 0.8 mL of PBS supplemented with 25 µg/mL chloramphenicol (to block protein translation), transferred into a 2 mL tube, and sonicated for 40 s, 20% power using the VCX130 Sonicator and CV18 tip (Sonics&Materials Inc., Newtown, CT, USA) on ice followed by centrifugation at 46,090× *g* for 4 min at 0 °C. A total of 100 µL of supernatant were mixed with 2.9 mL of PBS buffer supplemented with 25 µg/mL of chloramphenicol (pre-warmed at 37 °C for 10 min) in a 5 mL quartz cuvette. Fluorescence kinetics was further measured using the CM2203 spectrofluorometer (Solar, Minsk, Belarus) at 37 °C.

For preparative protein purification for X-ray crystallography, the bacterial cells expressing the LSSmScarlet protein with a His-tag and Tobacco Etch Virus (TEV) protease cleavage site were harvested by centrifugation for 20 min at 5000× *g* rpm and 4 °C using the Avanti J-E centrifuge (Beckman Coulter, USA). The pellet was further resuspended in 40 mM Tris(tris(hydroxymethyl)aminomethane)-HCl buffer, pH 7.5, supplemented with 400 mM NaCl, 10 mM Imidazole, 0.2% Triton X-100 and 1 mM PMSF (phenylmethylsulfonyl fluoride) (7 mL per 1 g of the cells) and disrupted by sonication (2 s pulse-6-s pause, 45% amplitude, for total time of 5 min). The crude cell extract was centrifuged for 30 min at 28,000× *g* and 4 °C using the Avanti J-E centrifuge (Beckman Coulter, USA). The supernatant was applied to a 5 mL Ni-NTA Superflow column (Qiagen, Hilden, Germany) equilibrated with the binding buffer (40 mM Tris-HCl, pH 7.5, containing 400 mM NaCl, 10 mM imidazole and 0.1% (*v*/*v*) Triton X-100). The column was future washed using the same binding buffer without Triton X-100 followed by a second washing using binding buffer in the absence of Triton X-100, supplemented with 40 mM imidazole. Protein elution was performed using the same binding buffer without Triton X-100 and supplemented with 300 mM imidazole. Both 1mM DTT (dithiothreitol) (final concentration) and 1mM EDTA (ethylenediaminetetraacetic acid) (final concentration) were added to the protein solution and mixed with TEV protease (1 mg per 10 mg of the protein). The final mix was dialyzed for 16 h in dialysis buffer (40 mM Tris-HCl, pH 7.8, 400 mM NaCl, 5 mM Imidazole, 2 mM BME (2-mercaptoethanol), 1 mM EDTA), at +4 °C (monitoring of the His-tag cleavage was conducted using 12% SDS-PAGE (sodium dodecyl sulphate–polyacrylamide gel electrophoresis)). After dialysis, the protein solution was applied to a Ni-NTA Superflow column (Qiagen, EU), equilibrated with dialysis buffer; TEV protease and cleaved His-tag were absorbed by the Ni-NTA Superflow column (Qiagen, EU), and the flow-through was concentrated until there was a 1 mL volume using a 10 kDa cutoff centrifugal filter device (Millipore, Burlington, MA, USA). The buffer of the concentrated protein was exchanged for the 20 mM Tris pH 7.8 buffer using the HiTrap Desalting 5 mL column (GE Healthcare, Sweden). The concentrated protein was further applied to a 1 mL ResourceQ column (GE Healthcare, Sweden) equilibrated with the same 20 mM Tris pH 7.8 buffer. Recombinant LSSmScarlet was eluted using a linear gradient from 0 to 1M NaCl; the protein was eluted by two peaks at 150 and 180 mM NaCl. Protein concentration was measured for each of the fractions individually using the Bicinchoninic Acid Protein Assay Kit (Sigma-Aldrich, Saint Louis, MO, USA), with the BSA (bovine serum albumin) protein standard (P0914-5AMP, Sigma-Aldrich, Saint Louis, MO, USA) solution as the standard. Total protein yield was 2.5 and 2.6 mg for the first and second fractions, respectively. The fractions were concentrated individually using a 10 kDa cutoff centrifugal filter device (Millipore, Burlington, MA, USA) until a 5 mg/mL concentration was reached and loaded onto the Superdex 75 10/300 GL column (GE Healthcare, Sweden), in a 20 mM Tris-HCl, pH 7.8, 200 mM NaCl buffer. In both cases, 11.5–13.5 mL fractions (corresponding to the monomer) were collected, and finally both fractions were combined and concentrated until a 10 mg/mL concentration for the crystallization was achieved. The total yield of the protein was 5 mg. Protein purity at each step was monitored using 12% SDS-PAGE. Chromatography was performed using ÄKTA prime plus and ÄKTA explorer 100 systems (GE Healthcare, Sweden).

### 3.3. Protein Crystallization

An initial crystallization screening of LSSmScarlet was performed with a robotic crystallization system (Rigaku, USA) and commercially available 96-well crystallization screens (Hampton Research and Anatrace, USA) at 15 °C using the sitting drop vapor diffusion method. The protein concentration was 10 mg/mL in the following buffer: 20 mM Tris-HCl, 200 mM NaCl, pH 7.8. Optimization of the initial conditions was performed by the hanging-drop vapor-diffusion method in 24-well VDX plates. Rod-like crystals were obtained within 2 weeks in the following conditions: 0.1 M Lithium sulfate, 0.1 M Ammonium acetate, 0.1 M Bis-tris pH 6.5, 23% PEG 3350.

### 3.4. Data Collection, Processing, Structure Solution and Refinement

LSSmScarlet crystals were briefly soaked in a 100% Paratone oil (Hampton research, USA) immediately prior to diffraction data collection and flash-frozen in liquid nitrogen. The X-ray data were collected from a single crystal at 100 K at the beamline “Belok-RSA” of the Kurchatov SNC (Moscow, Russia) [25]. The data were indexed, integrated and scaled using the Dials program [26] (Table 3). The program Pointless [27] suggested the C2 space group.

The structure was solved by the molecular replacement method using the MOLREP program [28] and the structure of the Red Fluorescent Protein mScarlet (PDB ID 5LK4) as an initial model. The refinement of the structure was carried out using the REFMAC5 program of the CCP4 suite [29]. The visual inspection of electron density maps and the manual rebuilding of the model were carried out using the COOT interactive graphics program [30]. The resolution was successively increased to 1.40 Å, and the hydrogen atoms in fixed positions, as well as anisotropic refinement, were introduced during the final refinement cycles. In the final model, an asymmetric unit contained one independent copy of the protein of 227 residues with the chromophore together with 272 water molecules, two calcium ions and one sulfate molecule from the crystallization solution. The first residue from the N-terminal, G226 as well as the last one residue from the C-terminal part of the protein were not visible in electron density.

### 3.5. Structure Analysis and Validation

The visual inspection of the structure was carried out using the COOT program and the PyMOL Molecular Graphics System, Version 1.9.0.0 (Schrödinger, USA). The structure comparison and superposition were made using the PDBeFold program [31], while contacts were analyzed using the PDBePISA [32] and WHATIF software [33].

### 3.6. Mammalian Plasmids Construction

In order to construct the pAAV-*CAG*-LSSRFP-P2A-EGFP plasmids, the LSSRFP genes were PCR amplified as the KpnI-AgeI fragments, using LSSCy-KpnI/LSSCy-AgeI-r or LSSCR-KpnI/LSSCR-AgeI-r primers listed in the Appendix A, and swapped with the R-GECO1 gene in the pAAV-*CAG*-R-GECO1-P2A-EGFP vector.

In order to construct the pLSSmScarlet-actin plasmid, the LSSmScarlet gene was PCR amplified as the NheI-HindIII fragment, using NheI-LSSmSc/LSSmSc-HindIII-r primers listed in Appendix A, and swapped with the TagBFP gene in the pTagBFP-actin vector (Evrogen, Russia).

In order to construct the pLSSmScarlet-tubulin plasmid, the LSSmScarlet gene was PCR amplified as the NheI-BsrGI fragment, using NheI-LSSmSc/LSSmSc-HindIII-r primers listed in Appendix A, and swapped with the TagGFP2 gene in the pTagGFP2-tubulin vector (Evrogen, Russia).

In order to construct the pAAV-CAG-H2B-LSSmScarlet plasmid, the LSSmScarlet gene was PCR amplified as the BglII-HindIII fragment, using mSc-BglII/mCherry-HindIII-r primers listed in Appendix A, and swapped with the B-GECO1 gene in the pAAV-CAG-H2B-B-GECO1 vector [7].

In order to construct the pAAV-CAG-dMito-LSSmScarlet plasmid, the LSSmScarlet gene was PCR amplified as the XhoI-EcoRI fragment, using LSSmSc-XhoI/mCherry-EcoRI-r primers listed in Appendix A, and swapped with the mCherry gene in the pAAV-CAG-dMito-mCherry vector.

In order to construct the pLU-CMV-vimentin-LSSmScarlet plasmid, the LSSmScarlet gene was PCR amplified as the BamHI-BsrGI fragment, using LSSmSc-BamHI/LSSmSc-XbaI-r primers listed in the Appendix A, and swapped with the mCherry gene in the pLU-CMV-vimentin-NeonOxIrr vector [20].

Vimentin-emiRFP703 plasmid was purchased from Addgene (#136566).

### 3.7. Mammalian Live- and Fixed-Cell Imaging

Transient transfection of the HeLa Kyoto cells was performed in a 24-well format using lipofectamine reagent according to the manufacture’s protocol. Cells were cultured using DMEM medium supplemented with 10% FBS, Glutamine, 50 U/mL penicillin and 50 U/mL streptomycin, at 37 degrees and 5% CO_2_. HeLa cell cultures were imaged 24 h after the transient transfection using a laser spinning-disk Andor XDi Technology Revolution multi-point confocal system (Andor Technology, UK) equipped with an inverted Nikon Eclipse Ti-E/B microscope (Nikon Instruments, Japan), a 75 W mercury-xenon lamp (Hamamatsu, Japan), a 60× oil immersion objective NA 1.4 (Nikon, Japan), a 16-bit Neo sCMOS camera (Andor Technology, UK), laser module Revolution 600 (Andor Technology, UK), spinning-disk module Yokogawa CSU-W1 (Andor Technology, UK). The green and lss-red fluorescence were acquired using 80% of the 488 nm laser power (14 µW/cm^2^ before objective lens), a confocal dichroic mirror 405/488/561/640 and filter wheel emission filters 525/50 and 617/73, respectively. During imaging, cells were incubated at 37 degrees and 5% CO_2_ using a cage incubator (Okolab, Italy).

Two-photon imaging of the cells was performed on a 33 mm dish with glass-bottom. Furthermore, 24 h after transfection in a 24-well format, cells were treated with 0.25% trypsin solution and transferred onto the 33 mm dish. Then, 24 h later, cells were imaged using a two-photon microscope system (Thorlabs, Newton, NJ, USA) equipped with the Chameleon Ultra I Ti:Sapphire laser (Coherent, Santa Clara, CA, USA), X-Cite 200DC light source (Lumen Dynamics, Canada), Galvo-Galvo Scan head and controller (Thorlabs, Newton, NJ, USA), the Model 302A amplifier (ConOptics, Danbury, CT, USA), 25× water dipping or immersion objective 1.10 NA (Nikon, Japan) and 525/50 and 607/70 emission filters.

### 3.8. Statistics

To estimate the significance of the difference between two values, we used the Mann–Whitney Rank Sum Test and provided *p*-values (throughout the text in the brackets) calculated for the two-tailed hypothesis. We considered the difference as significant if the *p* value was <0.05.

### 3.9. Molecular Modeling

The full-atom molecular model of the LSSmScarlet was built based on the X-ray structure obtained in this study. It was solvated in a rectangular water box of 80·62·68 Å^3^. The CHARMM27 [34,35] force field parameters for the protein and chromophore and TIP3P [36] parameters for water molecules were utilized for the preliminary equilibration of the system. We performed 1000 steps energy minimization followed by the 10 ns classical molecular dynamic (MD) run. Calculations were performed in the NAMD program package [37]. Thus, the prepared system was utilized for the molecular dynamics simulations with the combined quantum mechanics/molecular mechanics (QM/MM) potentials. The QM subsystem included the chromophore and the side chains of neighboring residues: R71, R96, D147, E149, R198, E216 and 4 water molecules of the chromophore-containing pocket. The QM subsystem was treated at the PBE0-D3/6-31G** level [38,39,40] on the ground electronic state energy surface, S_0_. Energies and forces in the QM subsystem were calculated with the TeraChem program [41]. After equilibration of the system for 20 ps, we performed the QM/MM MD simulation on the first excited singlet state, S_1_. We relied on the DFT functional CAM-B3LYP [42] with the def2-SVP basis set utilized for a similar system in ref [43]. Excited state QM/MM MD simulations were performed in the ORCA program [44]. All QM/MM MD simulations were performed with the help of the NAMD interface for the QM/MM treatment of the system [45]. All molecular dynamic simulations were performed at *p* = 1 atm and T = 300 K with the 1 fs integration time step.

### 3.10. Fixation of the Cells with 4% PFA

Before fixation, cells in a 24 well plate with glass-bottom were washed with 1 mL of DPBS buffer. Next, fixation of the cells was performed with 1 mL of 4% PFA for 15 min at room temperature. The sample was further washed with 1 mL of DPBS buffer and kept in 1 mL of DPBS buffer for further imaging.

### 3.11. STED Imaging

STED imaging was performed on a commercial STED microscope (STED Facility line, Abberior Instruments, Germany) equipped with 60×/1.42 N.A. The oil STED White objective lens (UPLXAPO60XO, Olympus, Japan) and standard 595 and 775 nm pulsed depletion lasers were used. The pixel size for the STED and confocal images was set to 30–40 nm. All images shown are raw data. Images were acquired using Imspector software (Max-Planck Innovation, Göttingen, Germany).

## 4. Conclusions

In conclusion, we developed and characterized a set of novel LSSRFPs characterized by the highest fluorescence brightness at a 488 nm excitation. LSSmScarlet was the first LSS version of the bright mScarlet red fluorescent protein (Figure 1). LSSmScarlet had medium brightness at a 488 nm excitation among a set of the developed LSSRFPs both as a purified protein (Table 1) and in the cytosol of mammalian cells (Figure 3b and Table 2). Moreover, LSSmScarlet had a moderate photostability and maturation rate compared to other LSSRFPs (Figure 2e,f and Table 1). However, LSSmScarlet was the only one among developed LSSRFPs for which fluorescence was not sensitive to alkaline changes of a pH up to 10 values (Figure 2d and Table 1); LSSmScarlet had also the highest pH-stability under acidic conditions. Furthermore, LSSmScarlet was monomeric (Figure 2c) and was successfully applied for the labeling and imaging of individual proteins in mammalian cells (Figure 4).

Since dCyRFP2s and dCyOFP2s and CRISPRed2s LSSRFPs proteins were developed from mCyRFP1, CyOFP1 and mCRISPRed LSSRFPs, respectively (Figure 1), we compared their biochemical properties to those of their parental proteins.

In contrast to monomeric mCyRFP1 LSSRFP, its derivative dCyRFP2s acquired a dimerization tendency due to the introduced mutations (Figure 2c). As compared to the mCyRFP1 progenitor, dCyRFP2s demonstrated 1.33- and 1.29-fold higher brightness at a 488 nm excitation as the purified protein (Table 1) and in mammalian cells (Figure 3b and Table 2), respectively. The dCyRFP2s variant had similar photostability and matured 1.5-fold slower compared to mCyRFP1 (Figure 2e,f and Table 1).

Similar to the CyOFP1 progenitor, dCyOFP2s was demonstrated to be a dimer (Figure 2c). dCyOFP2s had the highest brightness at a 488 nm excitation among all developed LSSRFPs both as purified proteins (Table 1) and in mammalian cells (Figure 3b and Table 2). dCyOFP2s also demonstrated the highest photostability at a 488 nm excitation and fastest maturation rate among all developed LSSRFPs (Figure 2e,f and Table 1).

In contrast to monomeric mCRISPRed LSSRFP, CRISPRed2s protein revealed a tendency to tetramerization (Figure 2c). CRISPRed2s protein had similar characteristics as mCRISPRed protein (Figure 2 and Table 1), except for a 3.9-fold faster maturation rate (Figure 2f). Similar to LSSmScarlet, CRISPRed2s had the highest pH stability at acidic conditions (Figure 2d).

Next, we solved the crystal structure of LSSmScarlet protein and suggested the mechanism of ESPT in LSSmScarlet protein, and elucidated the role of R198 residue in ESPT to the chromophore.

Finally, we demonstrated two- and three-color confocal and two-color 2P imaging with LSSmScarlet protein. We found that LSSmScarlet protein is not applicable for super-resolution STED microscopy as a consequence of the LSSRed-to-Red photoconversion with 775 nm depletion light; however, LSSRed-to-Red photoconversion was not efficient upon one-photon illumination with 405 and 468 nm light.

Overall, the new set of LSSRFPs that we present offers researchers a choice for the LSSRFP with the highest brightness, extreme pH-stability, high photostability, fast maturation or monomeric behavior, depending on the particular application. LSSmScarlet and CRISPRed2s due to their high pH-stability might be applied for the labeling of acidic organelles such as lysosomes. Monomeric behavior of LSSmScarlet allows using it for the labeling of subcellular structures, however, with limited applicability for the super-resolution STED microscopy; photoconversion of LSSmScarlet protein with 775 nm light makes it good candidate for photoactivation localization microscopy (PALM) imaging similar to the PSLSSmKate protein [22]. High brightness of dCyRFP2s and dCyOFP2s might be suitable for the labeling of cells and tissues, and for the development of the novel genetically encoded biosensors. Monomerized versions of the dCyRFP2s and dCyOFP2s LSSRFPs with high photostability might be possible candidates for super-resolution STED microscopy. Fast maturing dCyOFP2s might be useful for tracking promoter triggering and gene expression initiation.

## Figures and Tables

**Figure 1 ijms-22-12887-f001:**
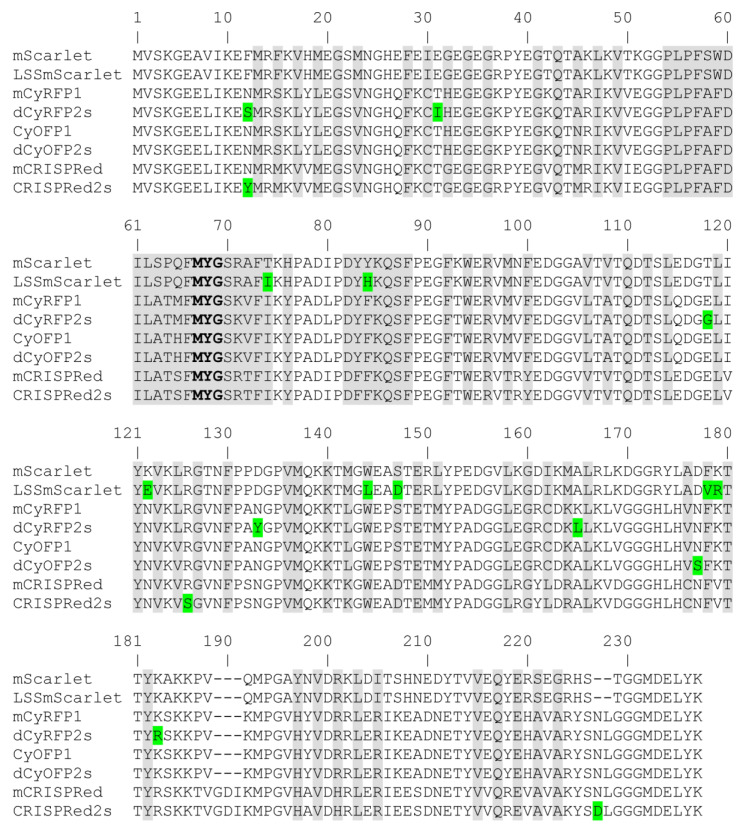
Alignment of the amino acid sequences for the developed LSSRFPs (LSSmScarlet, dCyRFP2s and dCyOFP2s) and their progenitors (mScarlet, mCyRFP1 and CyOFP1). Mutations in the developed LSSRFPs are highlighted in green color. Chromophore-forming residues MYG are selected in bold. Residues buried in the β-barrel are in grey.

**Figure 2 ijms-22-12887-f002:**
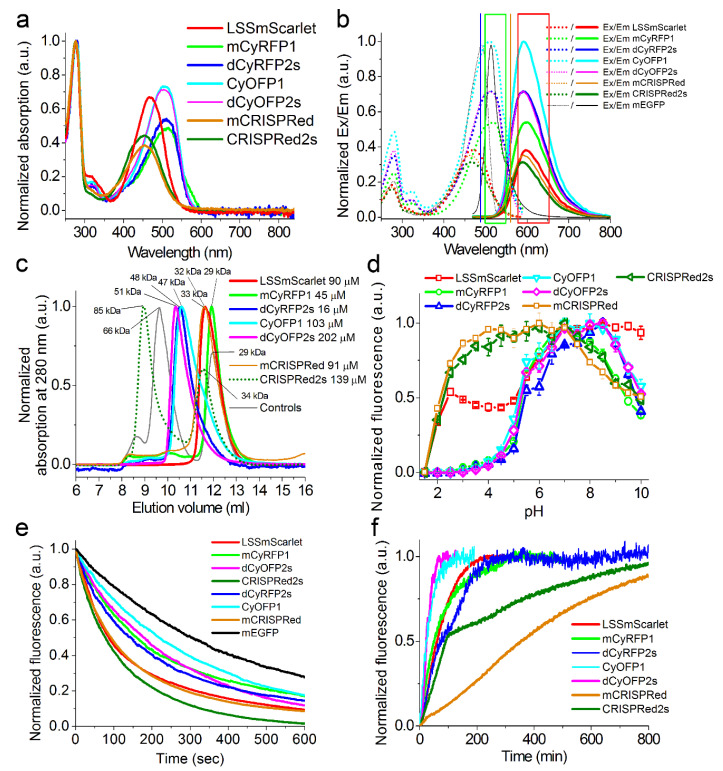
In vitro properties of the purified LSSRFPs proteins. (**a**) Absorption spectra for LSSRFPs proteins in PBS buffer at pH 7.40. (**b**) Excitation and emission spectra for LSSRFPs and mEGFP in PBS buffer at pH 7.40. The 488 and 561 nm excitation laser lines and green/red emission filters used for dual-color imaging are indicated as blue and orange vertical lines and green/red boxes, respectively. (**c**) Fast protein liquid chromatography of LSSRFPs proteins. LSSRFPs were eluted in 20 mM Tris-HCl (pH 7.80) and 200 mM NaCl buffer. The molecular weights of LSSRFPs were calculated from a linear regression of the dependence of logarithm of control molecular weights vs. elution volume (Appendix A). (**d**) Red fluorescence intensity for LSSRFPs as a function of pH. (**e**) Photobleaching of LSSRFPs and mEGFP proteins droplets in oil under continuous wide-field imaging using metal halide lamp (3.78 mW/cm^2^ light power before objective lens). (**f**) Maturation of LSSRFPs in PBS buffer at pH 7.40. (**d**–**f**) Three-six replicates were averaged for analysis. Error bars represent the standard deviation.

**Figure 3 ijms-22-12887-f003:**
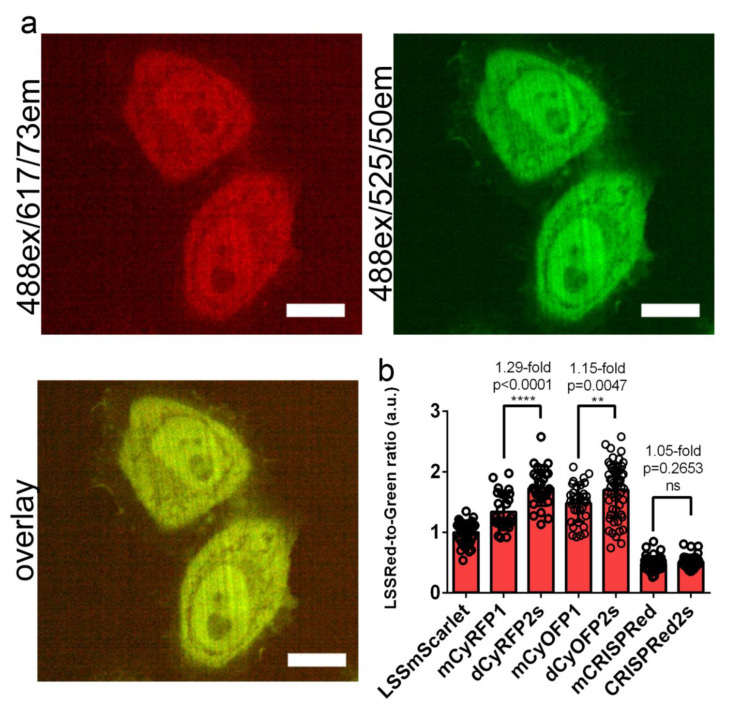
Comparison of the brightness of the LSSRFPs proteins in HeLa cells. (**a**) Confocal images of HeLa cells expressing the LSSmScarlet-P2A-EGFP protein (please, see Appendix A for localization images for other LSSRFPs). LSSRed (488ex and 617/73em) and green (488ex and 525/50em) fluorescence channels and their overlay are shown. Scale bar, 100 µm. (**b**) The averaged brightness for the LSSRFPs proteins in HeLa cells normalized to the brightness of the EGFP protein expressing in the same cells. The EGFP protein is connected to LSSRFPs via P2A-self cleavable linker. Error bars are standard deviations across twenty-eight–fifty-eight cells. *p* values show statistical difference between the respective values. Ns, not significant, *p* value is >0.05. ****, *p* value is <0.0001. **, *p* value is < 0.01.

**Figure 4 ijms-22-12887-f004:**
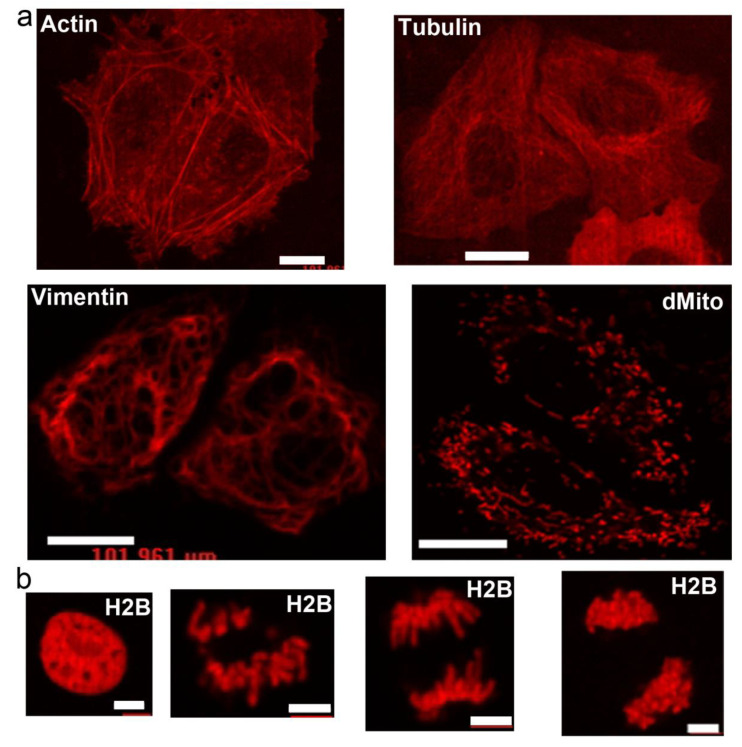
Localization of the LSSmScarlet protein in different fusions in mammalian cells. (**a**) Confocal images of HeLa cells expressing the actin-LSSmScarlet, tubulin-LSSmScarlet, vimentin-LSSmScarlet and dMito-LSSmScarlet fusions. Scale bar, 100 µm. (**b**) Confocal images of non-dividing and dividing HeLa cells expressing the H2B-LSSmScarlet fusion. Scale bar, 50 µm.

**Figure 5 ijms-22-12887-f005:**
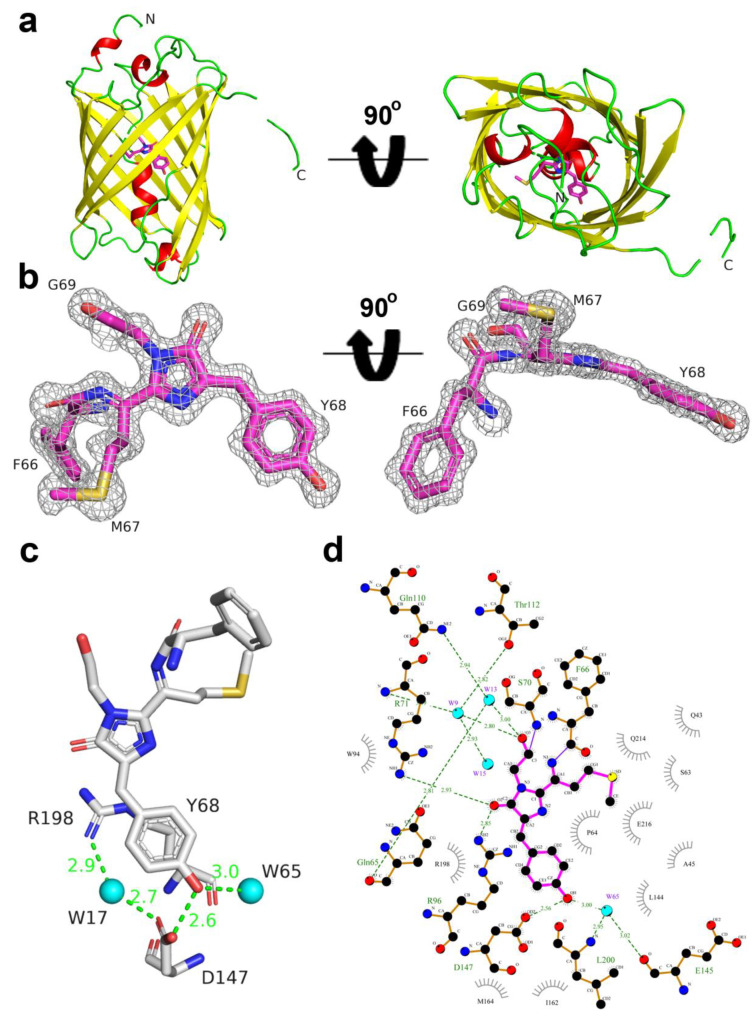
X-ray structure of the LSSmScarlet protein. (**a**) Cartoon representation of the overall LSSmScarlet structure. Chromophore, β-sheets, α-helixes and disordered structures are shown in pink, yellow, red and green colors, respectively. The orientation of the panel on the right is rotated 90° around the horizontal axis with respect to that on the left. (**b**) The chromophore of the LSSmScarlet protein refined into the electron density map. The 2F_0_-F_C_ electron density map for the LSSmScarlet chromophore structure (sticks in pink) is contoured at 1.5σ level and shown as gray mesh. The orientation of the chromophore on the right is rotated 90° around the horizontal axis with respect to that on the left. (**c**) Hydrogen bond network around the phenolic hydroxyl group of the LSSmScarlet chromophore. (**d**) The immediate environment of the LSSmScarlet chromophore.

**Figure 6 ijms-22-12887-f006:**
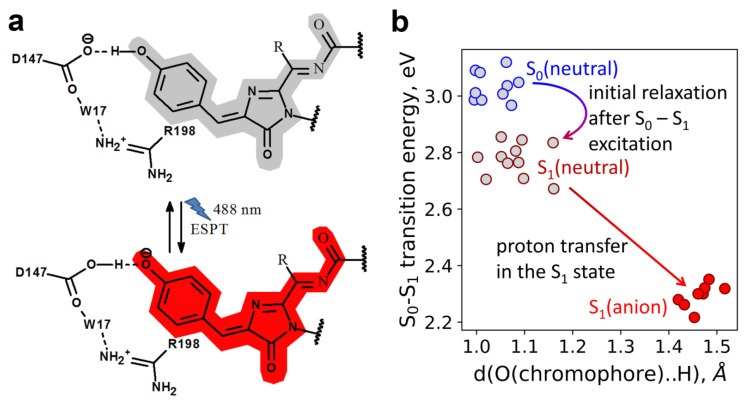
Suggested ESPT mechanism for LSSmScarlet. (**a**) The proton from the phenolic hydroxyl group of the chromophore migrates to the D147 residue after the excitation of the chromophore with 488 nm light. Upon relaxation into ground state from excited state proton comes from D147 residue back to the phenolic hydroxyl group of the chromophore. D147 residue forms water-mediated hydrogen bond with R198 residue (please, see also Figure 5c,d and Figure 7b). (**b**) The ESPT mechanism obtained in the QM/MM MD simulations. Circles correspond to the selected frames from trajectories: grey circles are for the neutral state (with blue and maroon edges for the S_0_ and S_1_ states, respectively), and red circles are for the anionic S_1_ state. The vertical axis depicts the S_0_–S_1_ transition energies calculated at the QM(CAM-B3LYP/def2-SVP)/MM(CHARMM) level.

**Figure 7 ijms-22-12887-f007:**
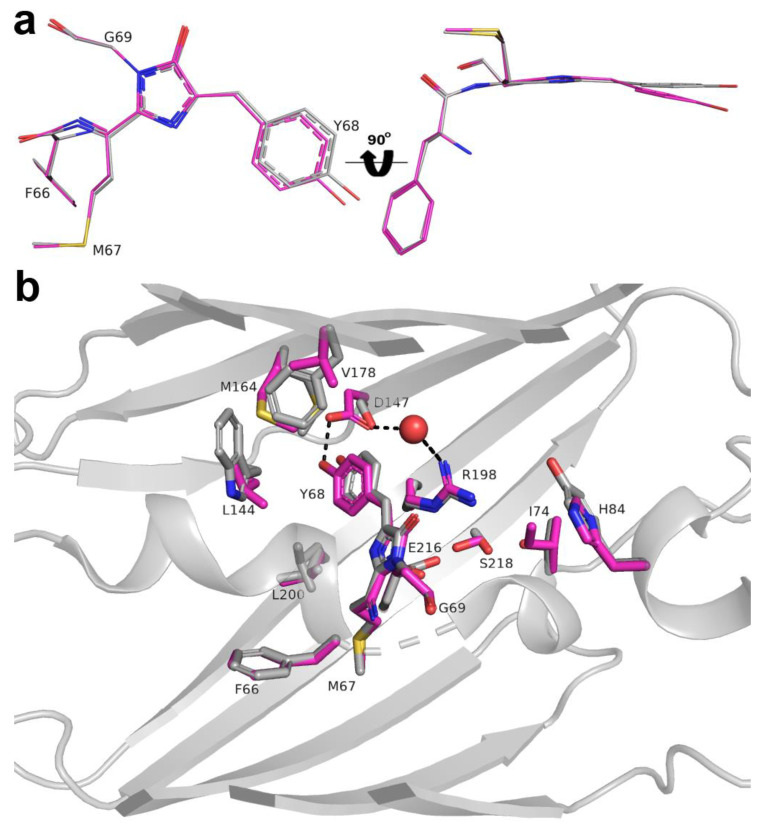
Structural comparison of the chromophores (**a**) and their immediate environments (**b**) for the LSSmScarlet and mScarlet (PDB: 5LK4) proteins. Water molecule (red sphere) provides hydrogen bonding (dashed lines) between D147 and R198 residues of the LSSmScarlet. D147 of LSSmScarlet makes a direct hydrogen bond (dashed line) with Y68 residue. Residues’ enumeration is shown for LSSmScarlet protein. In panel (**a**), the orientation of the chromophore on the right is rotated 90° around the horizontal axis with respect to that on the left.

**Figure 8 ijms-22-12887-f008:**
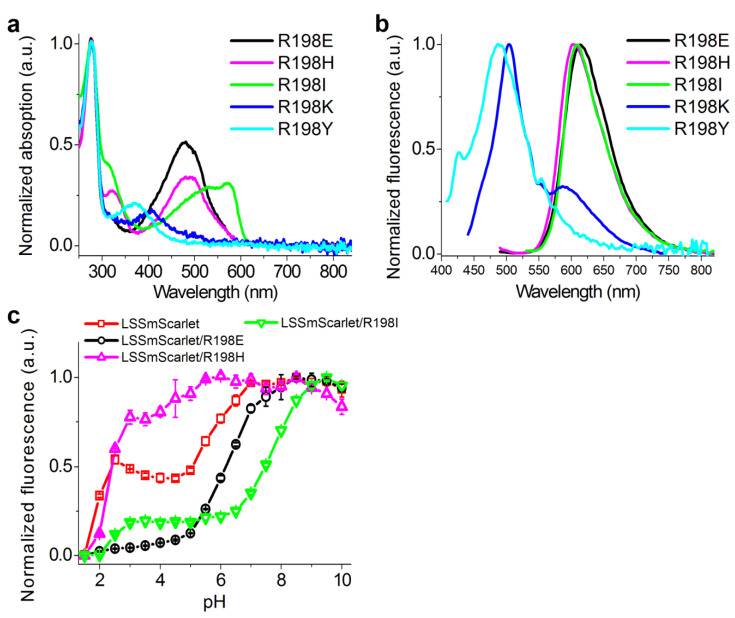
Spectral properties and pH stability for the purified LSSmScarlet mutants with substitutions at position 198. (**a**) Absorption spectra for LSSmScarlet mutants in PBS buffer at pH 7.40. (**b**) Emission spectra for LSSmScarlet mutants in PBS buffer at pH 7.40, excited at absorption maximum. (**c**) pH stability of LSSmScarlet and its mutants.

**Figure 9 ijms-22-12887-f009:**
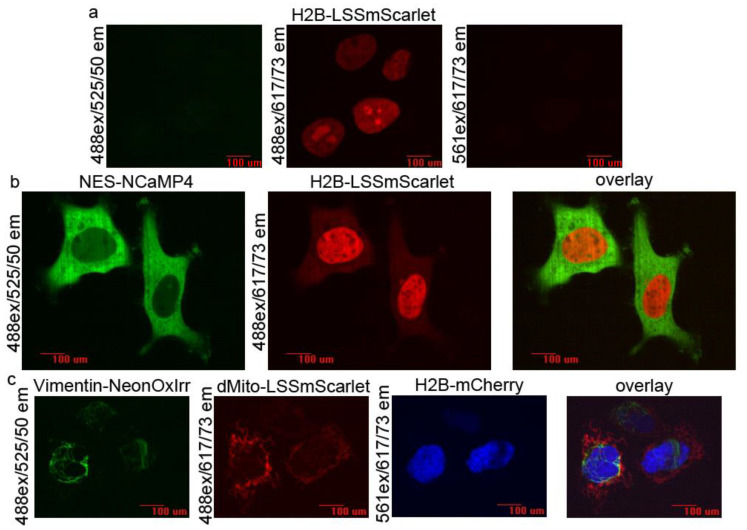
Two- and three-color confocal imaging with LSSmScarlet protein. (**a**) Confocal images of the HeLa cells expressing H2B-LSSmScarlet only in green (488ex/525/50em in green), LSSRed (488ex/617/73em in red) and red (561ex/617/73em in red) channels. We used the same exposure time and brightness/contrast adjustment levels for all channels. We do not see notable bleed-through of the LSSRFP fluorescence into both the green and red channels. (**b**) Two-color confocal imaging of the HeLa cells co-expressing NES (nuclear export signal)-NCaMP4 green calcium indicator and H2B-LSSmScarlet fusion LSSRFP in green (488ex/525/50em in green), LSSRed (488ex/617/73em in red) and overlayed channels using single-excitation with 488nm laser. (**c**) Three-color confocal imaging of the HeLa cells co-expressing the Vimentin-NeonOxIrr fusion green indicator for hydrogen peroxide, dMito-LSSmScarlet fusion LSSRFP and H2B-mCherry fusion RFP in green (488ex/525/50em in green), LSSRed (488ex/617/73em in red), red (561ex/617/73em in blue) and overlaid channels. Scale bar, 100 µM.

**Figure 10 ijms-22-12887-f010:**
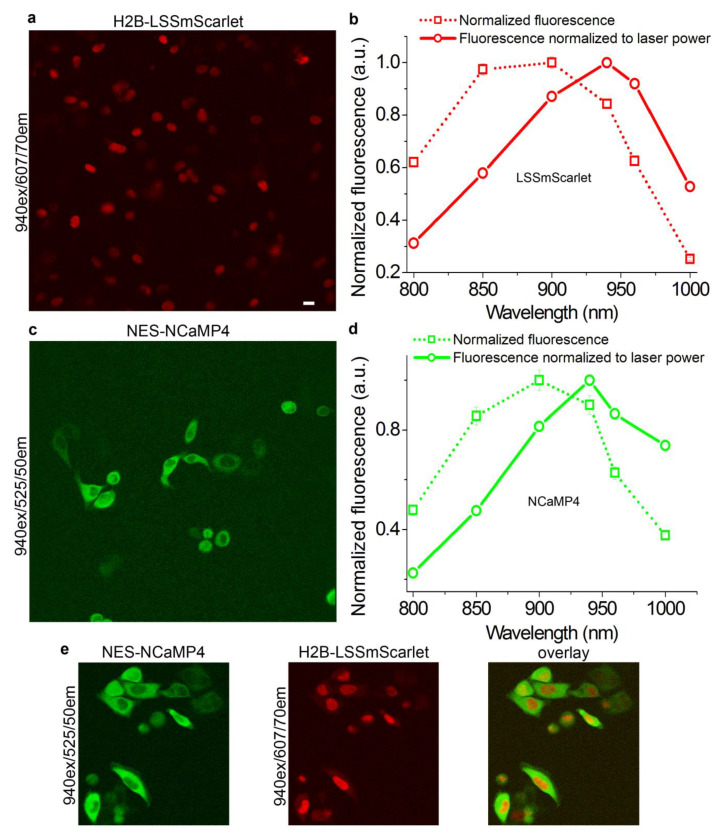
Simultaneous dual-emission two-photon imaging with LSSmScarlet and mNeonGreen-based calcium indicator expressed in HeLa cells. (**a**) Two-photon fluorescence image of the H2B-LSSmScarlet expressed in HeLa cells (940 nm excitation, 171 mW laser power; 607/70 emission filter). (**b**) Dependence of the H2B-LSSmScarlet fluorescence on two-photon excitation wavelength. (**c**) Two-photon fluorescence image of the NES-NCaMP4 green-yellow calcium indicator expressed in HeLa cells (940 nm excitation, 171 mW laser power; 525/50 emission filter). (**d**) Dependence of the NES-NCaMP4 fluorescence on two-photon excitation wavelength. (**e**) Two-photon fluorescence images of the H2B-LSSmScarlet and NES-NCaMP4 co-expressed in HeLa cells (940 nm excitation, 171 mW laser power). NES-NCaMP4 and H2B-LSSmScarlet were spectrally resolved using 525/50 and 607/70 emission filters, respectively. (**a**,**c**,**e**) Scale bars are the same for all images, 20 µm.

**Figure 11 ijms-22-12887-f011:**
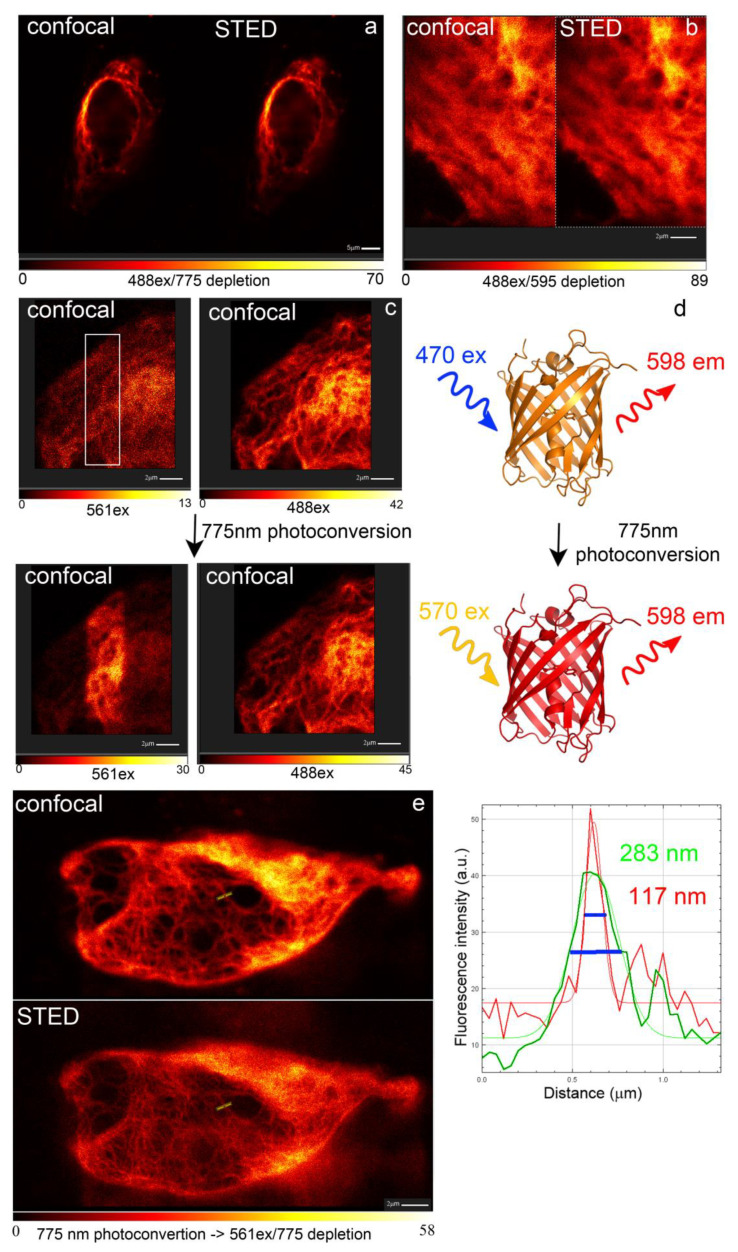
STED imaging of vimentin-LSSmScarlet fusion in HeLa cells. (**a**) Confocal and STED images of fixed cells expressing vimentin-LSSmScarlet. STED settings: 488 nm excitation—3.17% laser power; 775 nm STED—2% laser power; STED gated on delay 750 ps, width 8 ns; emission range of 560–725 nm; pixel dwell time—4 µs in 9 repetitions. (**b**) Confocal and STED images of fixed cells expressing vimentin-LSSmScarlet. STED settings: 488 nm excitation—5% laser power; 595 nm STED—100% laser power; on delay 750 ps, width 8 ns; emission range of 560–585 nm; dwell time—12 µs; line steps—4 repetitions, 16 total lines. (**c**) Confocal images of fixed cells expressing vimentin-LSSmScarlet before and after photoconversion with 775 nm depletion laser (50% power) in the region shown as a white box. (**d**) Schematic representation of excitation maxima of the LSSmScarlet protein as a consequence of the photoconversion with 775 nm light. (**e**) Confocal and STED images of live cells expressing vimentin-LSSmScarlet. The line profile for the marked area shows filament with 2.4-fold improvement of resolution. Line profile was fitted with y = a + (b − a) × exp(−(x − c) × (x − c)/2 × d × d)) equation. STED settings: 561 nm excitation—10.57% laser power; 775 nm STED—15% laser power; on delay 750 ps, width 8 ns; emission range of 570–765 nm; dwelltime—105 us; line steps—10 repetitions, 40 total lines.

**Table 1 ijms-22-12887-t001:** In vitro properties of the LSSRFPs. ^a^ Data from [3]. ^b^ QY was determined at pH 7.40. CyOFP1 (QY = 0.76 [3]) was used as the reference standard. ^c^ Extinction coefficients were determined by alkaline denaturation. ^d^ Molecular brightness was calculated as a product of the quantum yield and extinction coefficient at maximal absorption and normalized to the brightness of EGFP that has an extinction coefficient of 56,000 M^−1^·cm^−1^ and quantum yield of 0.6 [2]. ^e^ Brightness corrected for the decreased absorption at 488 nm. ^f^ Half-time to bleach until 50%. One-photon photobleaching was performed under metal halide lamp on droplets of protein solution (45 µM final protein concentration) in oil. Under identical conditions, mEGFP had photobleaching half-time of 305 ± 38 s. Standard deviations are shown.

Proteins	Abs, Ex/Em (nm)	QY (%) ^b^	ε (mM^−1^·cm^−1^) ^c^	Brightness vs. EGFP (%)		pK_a_	Monomeric State	Photobleaching Half-Time (s) ^f^	Maturation Half-Time (min)
				ex. at Max ^d^	ex. at 488 nm ^e^				
**LSSmScarlet**	466, 470/598	43 ± 2	30.2 ± 0.6	39	34	1.91 ± 0.01; 5.78 ± 0.06	Monomer	87 ± 16	61
**mCyRFP1**	514, 520/600	61.2 ± 0.4	30.6 ± 0.5	55	49	5.33 ± 0.03; 9.41 ± 0.12	Monomer	161 ± 26	51
**dCyRFP2s**	508, 516/592	58 ± 1	42 ± 2	73	65	5.44 ± 0.04; 9.817 ± 0.004	Dimer	144 ± 20	75
**CyOFP1**	503, 510/594	76 ^a^	45.1 ± 0.2	102	97	5.22 ± 0.07; >10	Dimer	223 ± 37	25
**dCyOFP2s**	503, 510/592	69 ± 7	36 ± 4	73	73	5.29 ± 0.03; >10	Dimer	176 ± 30	24
**mCRISPRed**	452, 462/592	42 ± 2	29.1 ± 0.9	36	27	2.14 ± 0.02 9.8 ± 0.3	Monomer	93 ± 7	354
**CRISPRed2s**	452, 464/590	38.2 ± 0.2	28.7 ± 0.9	32	24	2.23 ± 0.03 10.0 ± 0.1	Monomer-Tetramer	79 ± 13	90

**Table 2 ijms-22-12887-t002:** Brightness of LSSRFPs in HeLa cells at 488 nm excitation. ^a^ Brightness of LSSRFPs normalized to the brightness of the LSSmScarlet protein was estimated in HeLa cells relative to EGFP in LSSRFPs-P2A-EGFP fusion using the same 488 nm excitation light and 525/50 BP and 617/73 BP emission filters for EGFP and LSSRFPs, respectively. ^b^ Statistical significance of the difference between the normalized brightness for LSSmScarlet and the respective LSSRFP. NA, not applicable. ^c^ Brightness of the purified proteins corrected for decreased absorption at 488 nm excitation from Table 1 normalized to the brightness of LSSmScarlet.

Proteins	Normalized Brightness vs. EGFP in HeLa Cells ^a^ (*p* Value ^b^)	Normalized Brightness vs. EGFP for Pure protein ^c^	Difference in Brightness between Pure Protein and HeLa Cells, Fold
**LSSmScarlet**	1.00 ± 0.04 (NA)	1.00	0.93
**mCyRFP1**	1.34 ± 0.08 (<0.0001)	1.44	1.08
**dCyRFP2s**	1.73 ± 0.08 (<0.0001)	1.91	1.11
**CyOFP1**	1.48 ± 0.08 (<0.0001)	2.85	1.93
**dCyOFP2s**	1.70 ± 0.11 (<0.0001)	2.15	1.26
**mCRISPRed**	0.48 ± 0.03 (<0.0001)	1.00	2.08
**CRISPRed2s**	0.51 ± 0.03 (<0.0001)	0.71	1.38

**Table 3 ijms-22-12887-t003:** Data collection, processing and refinement. Values in parenthesis are for the highest-resolution shell.

**Data Collection**	
**Diffraction Source**	**“Belok-RSA” Beamline, NRC “Kurchatov Institute”**
Wavelength (Å)	0.75
Temperature (K)	100
Detector	CCD
Crystal-to-detector distance (mm)	120.00
Rotation range per image (°)	1.0
Total rotation range (°)	306
Space group	C2
*a*, *b*, *c* (Å)	84.52; 45.36; 59.04
α, β, γ (°)	90.0; 102.34; 90.0
Average mosaicity (°)	1.4
Unique reflections	42,242 (2137)
Resolution range (Å)	34.4–1.40 (1.42–1.40)
Completeness (%)	98.0 (100.0)
Average redundancy	6.0 (6.2)
〈*I*/σ(*I*)〉	7.2 (1.7)
Rmeas (%)	15.0 (67.2)
CC_1/2_	99.1 (81.3)
**Refinement**	
*R_fact_ (%)*	16.4
*R_free_ (%)*	19.0
Bonds (Å)	0.02
Angles (°)	2.16
*Ramachandran plot*	
Most favored (%)	98.6
Allowed (%)	1.4
*No. atoms*	
Protein	1850
Water	272
Chromophore	23
Sodium ion	2
Other ligands	5
*B-factors (Å* ^2^ *)*	
Protein	10.77
Water	21.34
Chromophore	7.06
Calcium ion	10.02
Other ligands	12.32

**Table 4 ijms-22-12887-t004:** In vitro properties of the LSSmScarlet and its mutants at 198 position. Minor peaks are shown in parentheses. Standard deviations are shown. NA, not applicable.

Proteins	Abs/Em (nm)	pK_a_
**LSSmScarlet**	466/598	1.91 ± 0.01; 5.78 ± 0.06
**LSSmScarlet/R198E**	480/614	6.181 ± 0.006
**LSSmScarlet/R198H**	487/604	2.403 ± 0.008
**LSSmScarlet/R198I**	571 (472)/610	7.76 ± 0.03
**LSSmScarlet/R198K**	405/503	NA
**LSSmScarlet/R198Y**	372/488	NA

## Data Availability

Data is contained within the article or Appendix A.

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
