# Peer review of "LSSmScarlet, dCyRFP2s, dCyOFP2s and CRISPRed2s, Genetically Encoded Red Fluorescent Proteins with a Large Stokes Shift"

_ijms, 2021, doi:10.3390/ijms222312887_

Round 1
Reviewer 1 Report
Dear editor,
I have read the manuscript describing the development of a Long Stokes Shift red fluorescent protein exhibiting an emission max over 100 nm longer than the excitation maxima. The work is nicely done, but I have two issues that need to be addressed before recommending publication.
The first is that there absolutely needs to be a figure showing the low emission of the new red FP in the 500-520 nm range in cells only transfected with LSSmScarlet. Currently, there is only co-transfection which does show nice green and red emission when excited at 488 nm. However, it is important to show the community whether this new FP will bleed into the green emission channel.
When comparing the ex/em spectra it is customary to plot normalized fluorescence, but that gives the impression that the proteins are the same brightness. That is clearly not the case. Please also plot the fluorescence emission normalized to the brightest probe in this study. This is important as the authors noted that the green FP used to as an internal control was o bright that it bled in to the red channel. Relative fluorescence matters.
The second issue deals with the certainty of the proton wire. While ESPT is very likely the cause of the LSS, I remain skeptical as to the exact nature of the proton transfer primarily because I do not see any evidence of two excitation states for LSSmScarlet. The R198I mutant does yield an absorbance peak at 570 nm which correlates to the original mScarlet FP but that is significantly less than the 280nm absorption peak. I’m guessing the fluorescence is very weak (again only normalized fluorescence is reported). I think this is excellent work, but I would ask the authors to suggest this mechanism as a strong possibility as opposed to the certainty they convey presently.
Minor comments
In results section 2.1 the authors state rational mutagenesis at positions 147, 162, and 164 followed by 8 rounds of random mutagenesis. What was the rationale for the S147D, I162D, and M164D mutations? The phrase ‘followed by eight rounds of random mutagenesis’ is confusing here. My understanding is that the 8 rounds of random mutagenesis was only performed on the S147D/M164C, the S147G/I164E and the S147D mutants.
I do not understand the rationale for choosing the T74I/Y84H/K122E/W144L/S147D/F178V/K179R mutant. What is meant by the highest ‘molecular brightness and the absence of red fluorescence’? I thought the goal was to get bright red fluorescence.
Why was only random mutagenesis done on CyOFP1, mCyRFP1 and mCRIPRed?
Table S1 wasn’t in the supplementary materials and the English of the table title is odd.
Supplemental
Figure S3 shows that all FPs tested yield green fluorescence when excited at 488. That would make them incompatible with coexpression of other GFPs.
Figure S4 needs a better explanation. In S4a it looks like the chromophore has been rotated and flipped (the same is true for chromophore in figure 5 as well). In S4b the legend states there is a hydrogen bond with E140, but E140 is not labelled.
Reviewer 2 Report
The study of. Subach et al. is aimed at adding novel fluorescent proteins to the palette of genetically-encoded probes for imaging of living cell morphology at single cellular and subcellular levels. The main finding of the paper is a new monomeric fluorescent protein LSSmScarlet with red-shifted emission spectra, which is suitable for fusion to produce composite constructs for intracellular targeting. Importantly, it provides bright and stable fluorescent labeling at far-red spectra using a standard 488 nm excitation, which is important for spectral separation though uncommon among conventional fluorescent proteins. This is a significant addition to currently available tools, which many researchers in the field of cell biology can benefit from. The experiments are well performed and described in many details. However, there are some flaws in data presentation and interpretation to address in the revised version of the manuscript.
Major
The authors claim that LSSmScarlet is monomeric so it is suitable for fusion with other proteins to resolve fine details of subcellular structures. However, they have not employed any high-resolution methods (such as STED, TURF of two-photon imaging) in live cells. What was the spatial resolution of the best optical method used in cells? Was it good enough to provide evidence of monomeric composition of the fused proteins in vivo and to resolve subcellular structures of interest?
Minor
1) Abstract
Mentioning of the suitability for two-photon microscopy in the very first sentence of the paper is quite misleading because the authors did not use two-photon excitation in their work. Neither of the fluorescent proteins was tested with this method. Therefore, it should be removed from there.
2) Fig 3, S3
Vertical stripes on green panels do not look like something of biological origin. Is it an artifact of the scanning or sample preparation?
3) The authors should also have compared the properties of their probes to those of the other novel red proteins proposed for fusion and subcellular structural imaging (Matlashov et al., 2020 Nat Commun etc).
Reviewer 3 Report
Review Report
In the manuscript, the authors propose a mScarlet-based variant of a red fluorescent protein with a large Stokes shift and improved variants of other LSSRFPs obtained using both rational design and directed evolution approaches. The panel of the existing LSSRFPs is represented mostly by rather dim and dimeric proteins. Therefore, the development of bright monomeric red fluorescent proteins with a large Stokes shift is required.
Overall, this paper demonstrates interesting results and would deserve publication in IJMS after revisions.
- The authors used sfGFP to assist the folding of mutants, however, no reference is cited. It is advisable to add an appropriate citation.
- In Figure 2, the authors provide the spectral characteristics, pH dependence, photostability, and maturation rate of the new LSSRFPs. However, it would be more convenient for a reader to see how they are different from EGFP. Thus, I suggest including the EGFP excitation and emission spectra in the Figure. In addition, in Figure 2b please indicate the 488 nm (as a vertical line), excitation, and green/red emission filters used for dual-color imaging.
- Similarly, I suggest comparing the photostability of the newly obtained proteins with EGFP and add corresponding data in Figure 2e. This information could be used for the selection of optimal fluorescent proteins for continuous dual-color imaging.
- In Figure 2d, the authors show pH dependence for LSSmScarlet, which looks unusual. Could you please explain why LSSmScarlet fluorescence almost does not change in the pH range 2.5-5.0 and 7-10? Is it connected to two different forms of chromophore or something else? Could you please provide excitation and emission spectra of LLSmScarlet at different pH?
- In Figure 2a, we see that LSSmScarlet, CyOFP1, or CyOFP2s have similar absorption maximum values, but different extinction coefficients (EC, Table 1). I assume that this fact reflects different chromophore maturation efficiency. CyOFP1 EC is 1.5 higher than LSSmScarlet EC, but normalized to 280 nm (protein concentration) absorption spectra do not demonstrate 1.5 higher absorption maximum value, so probably CyOFP1 has lower maturation efficiency than LSSmScarlet. Could you please provide the table with maturation efficiency (percentage of proteins with properly folded chromophore) values for all proteins?
- There is a typo on page 3 section 2.2, first sentence: “pogenitor”.
- Reference [7] does not support the sentence where used.
- In Figures 3a and 3S, there are overlay pictures. Since all of the proteins do not have specific localization sequences, the fluorescence signal is distributed almost equally across the cells. What was the purpose of the overlaying green and red images?
- In Figure 4 could you please provide images with better resolution? The LSSmScarlet-vimentin image looks like a print screen with a few numbers at the bottom of the picture. LSSmScarlet-tubulin looks very blurry. I suggest bringing pictures to the same size, transfer captions (Actin, vimentin, dMito, and others) into the figure legend, and make scale bars the same width.
- In Figure 8, could you please change the color of the pH dependence curve for LSSmScarlet R198H and R198E to magenta and black, respectively?
- Is it possible to use proposed proteins with bright green fluorescent proteins? Have you noticed GFP redding during continuous dual-color imaging and, if so, did it hamper the imaging?
- In your opinion, which developed fluorescent proteins are more suitable for which applications?

Round 2
Reviewer 1 Report
The authors have addressed my concerns quite nicely.
Reviewer 3 Report
Dear Authors,
Thank you for the complete and extended answer to all questions. The article has become more prosperous and understandable, and new experimental data in the main text and supplementary materials made it more exciting and informative.